

# Collecting whales: processes and biases in Nordic museum collections

Lene Liebe Delsett

Department of Archaeology, Conservation and History, University of Oslo, Oslo, Norway

## ABSTRACT

Whales are unique museum objects that have entered collections in many ways and for different reasons. This work studies three Nordic natural history museum collections in Norway and Denmark with more than 2,500 whale specimens in total, and gathers the available biological and collection data on the specimens, which include skeletal elements, foetuses and organs preserved in ethanol or formalin, and a few dry-preserved organs. It finds that influx of specimens, which were mainly locally common species that were hunted, to the collections, mainly happened in the latest 1800s and earliest 1900s, fuelled by research trends, nation building, local whaling, and colonial mechanisms. Norway was a major whaling nation, but the largest hunt for whales in the Southern Ocean in the mid-1900s is not reflected in the Norwegian museum collections, probably because of the commercial focus of the whaling industry and logistical challenges, combined with limited research interest in zoological specimens at that time. The results demonstrate that it is important to understand these processes and the resulting biases for future research, outreach, and conservation.

# INTRODUCTION

Whales have always been important to humans, as a natural resource, as a research subject and in many cultures for hundreds of years (*Burnett, 2012*; *Gatesy et al., 2013*). The modern whaling that started in the late 19[th] century and later developed into large scale industrial whaling changed this relationship, and the hunt for whales from the 1890s to the 1980s, mainly for production of fats used in food, medicines, and machinery, has been called "the largest hunt in history" (*Rocha, Clapham & Ivashchenko, 2014*; *Tønnesen & Johnsen, 1982*). From 1900 to 1999, 2.9 million large whales (baleen and sperm whales) were killed, a reduction of up to 99–100% for some populations (*Rocha, Clapham & Ivashchenko, 2014*; *Roman & Palumbi, 2003*). After the cessation of large-scale whaling, globally from 1986 due of the moratorium set by the International Whaling Commission, some populations are showing signs of recovery, whereas others are not, and some are responding negatively to other anthropogenic pressures (*Albouy et al., 2020*; *Edwards et al., 2015*; *Heide-Jørgensen et al., 2013*; *Reeves et al., 2014*; *Savoca et al., 2021*). Today, five whale species are classified as critically endangered and twelve as endangered, in addition to several subspecies and subpopulations, and for many species, the status is unknown

Corresponding author
Lene Liebe Delsett,
l.l.delsett@nhm.uio.no

because of missing knowledge (*SSC Cetacean Specialist Group I, 2022*). A few countries were responsible for most of the whaling both regarding income, technological innovations, and skilled labour (*Rocha, Clapham & Ivashchenko, 2014*; *Schladitz, 2014*; *Tønnesen & Johnsen, 1982*). Among these are Russia and Japan, but also Norway, a small country that came to dominate global whaling from the late 1800s, but also had a long tradition of smaller scale subsistence whaling (*Kalland, 2014*; *Stein, 1994*; *Tønnesen & Johnsen, 1982*).

Today, whales are still important to humans, not just as a resource, at least at the same scale, but as part of ocean ecosystems, as local food resources, and as symbols of biodiversity, for cultural practices and religion. On the other hand, new economic practices have developed from the interest in whales, and the whale watching industry has been growing for several decades (*Suárez-Rojas, González Hernández & León, 2023*), cetaceans are held in captivity for entertainment, and new TV documentaries on whales are constantly being made. Not of least importance, people encounter whales in natural history museums, as whales are centrepiece objects in exhibitions all over the world, for instance in London, Singapore and Gothenburg (*Brito, Vieira & Freitas, 2019*; *Delsett & Spring, 2023*).

The exhibited specimens only represent a fraction of the specimens held by museums, as they usually have larger collections aimed at education and research purposes (Fig. 1) (*Pyenson, 2017*). Whales appear as different types of objects in natural and cultural history museums, in art or as part of religious practices, as a material in human-made objects, or documented in photographs and in writing (*Vieira et al., 2020*). This work however, concerns biological specimens in natural history museums, most commonly preserved as skeletons or in liquid in glass jars. These museums are again experiencing increased research focus towards the use of collection specimens for understanding among other aspects changing biodiversity patterns, and spread of diseases and toxins, using a range of methods (reviewed in *e.g.*, *Bakker et al., 2020*; *Hilton, Watkins-Colwell & Huber, 2021*; *Meineke et al., 2019*). Because whales will not be targeted by humans on a large scale in the foreseeable future and some are endangered, many specimens in museum collections are unique objects that will not be replaced.

However, using historically collected specimens in modern research is not straightforward, and specimens are not suitable for all types of research, because of inherent collection bias (*Bakker et al., 2020*; *Boakes et al., 2010*; *Pyke & Ehrlich, 2010*; *Uhen & Pyenson, 2007*; *Wehi, Whaanga & Trewick, 2012*). Because museum collections have been built up over hundreds of years, the aim, strategy and what is economically and logistically possible, as well as which objects are valued, has shifted repeatedly (*Bakker et al., 2020*). Using a framework on collection bias, one can imagine nature going through a series of filters, where only a fraction of the original biodiversity passes through each one, because bias either from natural or anthropologic causes is introduced at all steps in the process (*Uhen & Pyenson, 2007*; *Whitaker & Kimmig, 2020*). The resulting collections will thus not represent the actual whale population geography, demography or anatomy one to

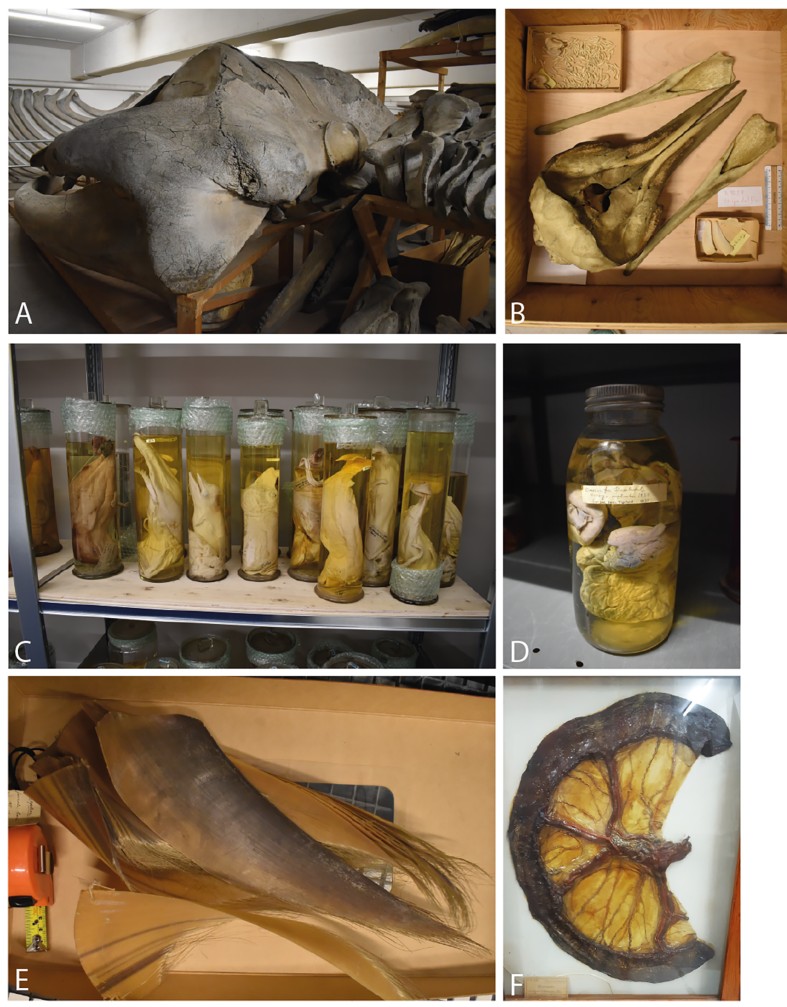

**Figure 1 Examples of whale specimens in museum collections.** (A) Blue whale (CN13), NHM Copenhagen. Osteological specimen, complete. (B) Striped dolphin (BM 9229), NHM Bergen. Osteological specimen, complete. (C) Wet collection, NHM Bergen. Foetuses, mostly minke whale. (D) Ovaries from fin whale in wet collection (NHMO-DMA-29084/1-P), NHM Oslo. (E) Baleen, NHM Oslo. (F) Dry preserved organ, NHM Copenhagen.

one, but instead be biased. Sometimes these biases can be systematic, whereas in other cases they are more random. Understanding collection history is thus vital for understanding what the collections represent and how they are biased, and what they can contribute in terms of scientific results (*Pyke & Ehrlich, 2010*), as they often represent the most comprehensive data available, despite the biases (*Boakes et al., 2010*).

Collection bias should be investigated for different taxonomical groups in order to be most precisely understood (*Benton et al., 2011*). This work uses three Nordic museum collections of whale specimens to map and discuss the collection processes. The aims are to analyse datasets for the whale specimens in the three collections including both biological data as well as data on how the specimens were collected; to discuss the collection processes

                                    

and sampling regime and how this has affected the resulting collections; and highlight possible biases. It also asks how the collections were impacted by whaling, and whether large number of whales that were killed is reflected in museum collections.

## Historical context

Collecting natural history specimens is largely influenced by history (*Anderson & Pietsch, 1997*). In 1814, Norway got its own constitution and entered a loose union with Sweden after being part of Denmark for several hundred years. In the last decades of the 1800s, Norwegian national identity grew, with increased demand for national institutions and full independence, which happened in 1905. Denmark had also expanded in Greenland since the mid-1700s. In 1953, Greenland went from being a colony to a part of the Danish nation, and in 1979 home rule was granted, and expanded in 2008 (*Gabriel, 2009*).

Whales have been hunted along the coasts of Europe and in the Arctic for hundreds of years. The whaling in the Arctic and North Atlantic from the 1600s largely affected bowhead whales and the Atlantic right whale, and the populations became severely reduced (*Cerca et al., 2022*; *Moore et al., 2021*; *Tønnesen & Johnsen, 1982*). From the late 1800s onwards, Norway became the world's dominant whaling nation. This started in 1864 when sealer Svend Foyn invented the steam-powered whale catcher and the exploding harpoon gun, and improved the on-shore whale processing, inventing modern whaling (*Burnett, 2012*; *Tønnesen & Johnsen, 1982*). This made it possible to catch the rorquals, fast-swimming baleen whales that did not float after death, and made whaling far more efficient. Consequently, it shifted the focus of the targeted species and changed the impacts on whale populations and their ecosystems. Geographically, this whaling started in northernmost Norway, along the coast of the Finnmark county and northwards in the Barents Sea, dominated by Svend Foyn's company. In this period, at least 18,000 whales were caught (blue, sei, fin and humpback whales) (*Davis, Gallman & Gleiter, 1997*; *Ringstad, 2011*; *Tønnesen & Johnsen, 1982*; *Øien, 2010*), with the largest catches made in the 1890s (*Tønnesen & Johnsen, 1982*). Whaling increasingly took place further from the shore because the local populations were reduced, and in 1904, whaling was prohibited in Northern Norway (*Tønnesen & Johnsen, 1982*). The whaling industry then turned to other hunting grounds, first in the Northern Hemisphere (Iceland, Faroe Islands and Newfoundland). After depleting the populations there, industrial whaling moved to the Southern Hemisphere, along the southern African coast and in the Antarctic. The whaling station in Grytviken at South Georgia was the main hub until full pelagic catch and processing made shore-based stations unnecessary (*Rocha, Clapham & Ivashchenko, 2014*; *Sanger & Dickinson, 1997*; *Tønnesen & Johnsen, 1982*).

The largest numbers of whales were killed in the last decades before bans on whaling, between 1950 and 1970. Norway had the largest fleet, and provided technology, knowledge transfer and skilled labour to other nations (*Schladitz, 2014*; *Tønnesen & Johnsen, 1982*). This industry provided large incomes to Norwegian actors. In 1957, "whale oil" had a value of 300 mill NOK—approximately half the value of the fisheries. Increasingly stricter

international regulations on whaling were developed from 1931 to 1982 (*Rocha, Clapham & Ivashchenko, 2014*). Norway objected repeatedly to these but ceased whaling in the Southern Hemisphere in 1961. Minke whales are still hunted in small numbers along the Norwegian coast. In addition to the industrial hunt for the large whales, there was a continuous hunt for toothed whales along the Norwegian coast (*Kalland, 2014*). Despite the scale and importance of Norwegian whaling, little research is done on its relation to museum collections, which is what this paper is addressing.

## MATERIALS AND METHODS

Three Nordic natural history museum collections of whales were studied in this work. The Natural History Museum collections in Oslo and Bergen (hereafter NHM Oslo and NHM Bergen, respectively) were the places and collections selected because they are the largest in Norway, and the one in Copenhagen, Denmark (hereafter NHM Copenhagen) is larger and serves as a comparison to the former two. Datasets of the whale specimens present in the three collections as of 2022 were assembled, and are used for the present results and discussion. The basis of these datasets are the museum databases, supplemented with other sources and personal observation, which included photo and recording of physical specimen labels during collection visits by LLD in April 2022–January 2023 to all three collections. See below for details on sources for information for each museum.

Whale specimens in the dataset are those in the zoological collections, and include skeletons, foetuses and organs preserved in ethanol or formalin, and dry preserved organs (Fig. 1), whereas cultural objects made from whales, casts, pictures, and drawings are not part of this study. All specimens were treated as separate entries, regardless of preservation technique and completeness. This means that more than one specimen might derive from the same individual, *e.g.*, if the skeleton is preserved in the osteological collection, and the inner organs in the wet collection. Subfossil specimens, which is a separate collection at NHM Bergen, were excluded. The museums' taxonomic assignments were not evaluated, but nomenclature was updated to follow the Society for Marine Mammalogy (*Committee on Taxonomy, 2022*). Data recorded for each specimen were: species, ontogenetic stage, sex, geographic location, collection year (see definition below), name and role of the collector, and how the specimen was acquired by the museum (donation, purchase, exchange or research project). Preparation types were recorded as: OT osteological specimens, including teeth (some specimens have dried soft tissue attached); B baleen; WF complete foetus or small juvenile with soft tissue, stored in ethanol or formalin; WO organs or stomach content stored in ethanol or formalin or DO organs, preserved as dry specimens.

At NHM Oslo, the basis of the dataset were the data extracted from the museum database (Corema) for all whale specimens, by head engineer Lars Erik Johannessen. This database includes both biological and collection data. However, many specimens lack data, and to fill these gaps, the following sources were used by LLD: physical specimen labels,

digitized acquisition books ("inntaksjournal"), gift registration book 1870–1879 ("gaveprotokoll"), collection registration book 1–5,000 ("samlingsjournal"), card catalogue ("kartotekkort"), and *Collett (1911–1912)*. According to the museum database, there are 317 whale specimens, and of these 196 were found and matched with the database. In addition, 188 specimens were found in the collection that could not be matched, often because of missing labels. At least 60 out of these 188 specimens are not registered at all. Eighty-four out of the 188 specimens, for which genus was unknown, were removed from the analysis. Most of these also lack other data. This left 421 specimens for analysis from NHM Oslo. The analysis is thus biased by collection management: some specimens might be counted twice, whereas others are not included. For the timeline, specimens registered for 1820 and 1834 are left out, as the oldest specimen that was found is from 1839.

At NHM Bergen the basis for the dataset were data on all whale specimens registered in the osteological, wet and exhibition collections, extracted from the museum database by curators Hanneke Meijer and Terje Lislevand. To fill in the missing information, the following sources were used by LLD: physical specimen labels, physical acquisition books for the period 1889–1997, digitized acquisition lists, *Collett (1911–1912)* and *Kalland (2014)*. At NHM Bergen, 267 out of 509 specimens were personally inspected by LLD in 2022. A total of 194 remaining specimens belong to the wet collection, which is recently inventoried and well-organized, and were included in this analysis. The 49 last specimens are present in the database but were not observed during the visits in 2022. Most likely these are present in the collection, and they are included in the analysis, even if some of them might have been discarded, introducing some bias.

For NHM Copenhagen, only the osteological collection is included, for practical reasons. The museum does not have a digital specimen database, and the dataset is assembled by personal observation by LLD in June 2022. The osteological collection at NHM Copenhagen contains 1,780 whale specimens. Out of these 36 were lacking information on genus or species, leaving 1,744 specimens for analysis.

The resulting datasets are openly shared as Supplemental Material in the process of producing open science, and it is hoped that they can be used for future research. Museum specimen numbers have several different abbreviations: BM, B, ZMUB and ZU at NHM Bergen; NHMO-DMA and M at NHM Oslo; and CN, MCE, M, FM at NHM Copenhagen. All three museums also have several specimens marked with museum numbers without letters, but that are clearly associated with the collections where they are physically located.

The study also had to deal with missing data, as this study collected many types of data from an extended time span, with a biased collection history, and many specimens miss some or all accompanying information. The recorded and missing information for the categories geography, collection year, sex and ontogeny is summarized in Table 1. It was ensured that corresponding values are compared, *e.g.*, only specimens with known collection year were compared when discussing temporal trends (Figs. 2 and 3). Regarding collection year, the year mentioned on the label or in the database sometimes refers to the actual collection, and in other instances to inclusion in the museum collection. In most

**Table 1 Available information for whale specimens in the three collections.**

| | | NHM Bergen | | NHM Oslo | | NHM Copenhagen |
|---|---|---|---|---|---|---|
| | | WF+ WO | B+OT+DO | WF+ WO | B+OT | OT |
| Geography | Recorded | 295 | 135 | 76 | 225 | 303 |
| | Not recorded | 31 | 45 | 29 | 91 | 1,479 |
| Collection year | Recorded | 257 | 120 | 86 | 246 | 206 |
| | Not recorded | 71 | 61 | 19 | 70 | 1,576 |
| Sex | Recorded | 76 | 64 | 14 | 72 | 457 |
| | Not recorded | 252 | 117 | 91 | 242 | 1,325 |
| Ontogeny | Recorded | | | | | |
| | Foetus | 311 | 3 | 66 | 5 | 11 |
| | Neonatal | 0 | 0 | 0 | 0 | 1 |
| | Juvenile | 3 | 7 | 0 | 18 | 13 |
| | Adult | 0 | 0 | 0 | 2 | 0 |
| | Not recorded | 179 | | 330 | | 1,719 |

**Note:**
Recorded and missing information about geography (country or ocean), collection year (see Material and methods for definition), sex and ontogenetic stage, organized by museum and wet (WF and WO) or dry (B, OT and DO) collection. With regard to ontogenetic stage, among the many "not recorded" specimens are likely a majority adult specimens. The category for juveniles include specimens recorded as "subadult".

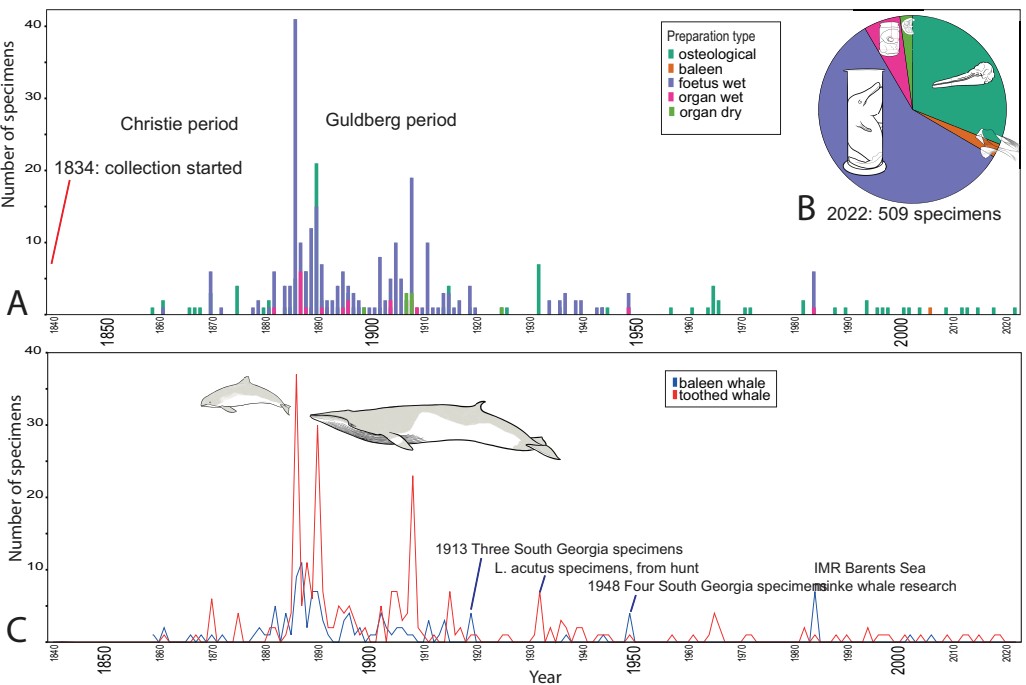

**Figure 2 Temporal trends for whale specimens entering the collection, NHM Bergen, 1840–2021.**
(A) Preparation type. (B) Distribution of preparation types 2022 (509 specimens). (C) Baleen and toothed whale collection. Peaks in 1881–1905 represent a large influx of *Phocoena phocoena*, but also *L. albirostris*, *L. acutus* and *O. orca*. Baleen whales are most commonly *B. acutorostrata*. The rare events of Southern hemisphere whaling related specimens are also recorded. Whale drawings: Nicola Dahle.

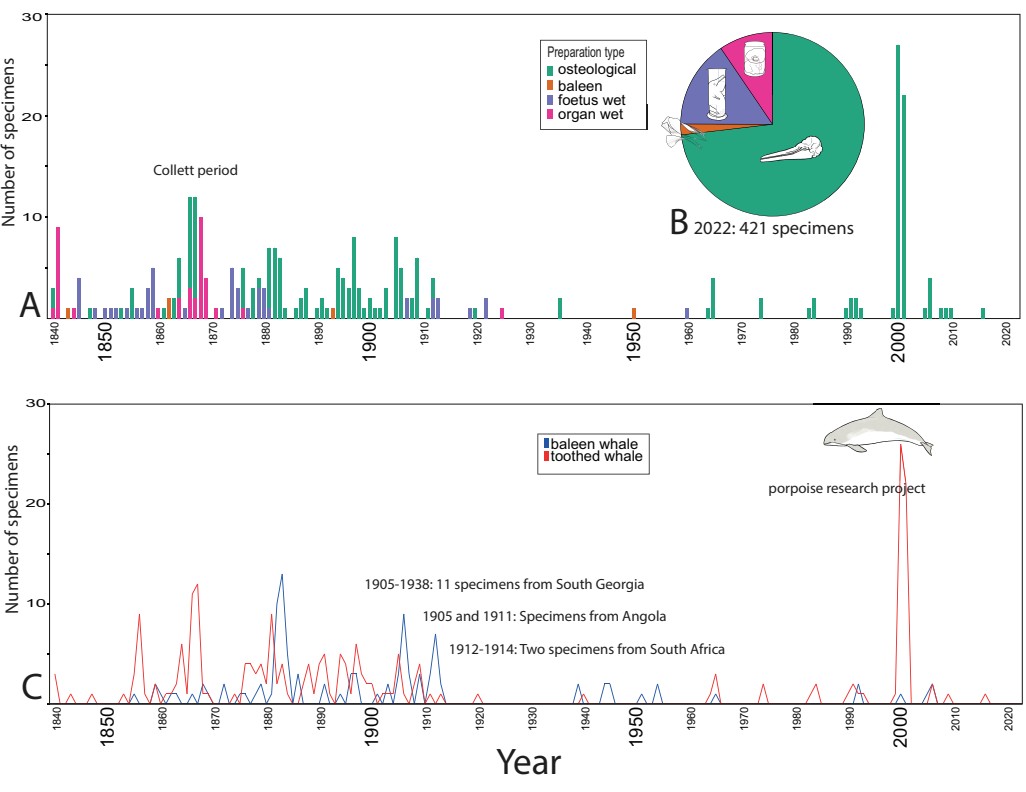

**Figure 3 Temporal trends for whale specimens entering the collection, NHM Oslo, 1840–2021.** (A) Preparation type. (B) Distribution of preparation types 2022 (421 specimens). (C) Baleen and toothed whale collection. Some of the Southern hemisphere whaling related specimens are also recorded, in addition to an explanation on the peak in year 2000. Whale drawings: Nicola Dahle.

cases, these lie within 1–3 years of each other, and the oldest value was always used. Most specimens are not labelled with ontogenetic stage, and many of these are assumed to be adult (Table 1). At NHM Bergen and NHM Oslo, most specimens for which ontogenetic stage is recorded, are foetuses in the wet collection. At NHM Copenhagen, ontogenetic stage is provided for 25 non-adult specimens. The sex of most whale specimens is also unknown. At NHM Bergen, among the 130 specimens where sex is registered, were 70 male and 60 female (54:46 ratio). At NHM Oslo and NHM Copenhagen, the majority of specimens for which sex is known, are harbour porpoise (*Phocoena phocoena*) osteological material.

For discussion on geographical origin of the whales (Fig. 4), the species compositions in the collections were compared to species occurring in Norway (including Svalbard, Jan Mayen and the Barents Sea); *Balaenoptera musculus, B. physalus, B. acutorostrata, B. borealis, Megaptera novaengliae, Balaena mysticetus, Eubalaena glacialis, Orcinus orca, Phocoena phocoena, Monodon monoceros, Physeter macrocephalus, Globicephala melas, Delphinapterus leucas, Hyperoodon ampullatus, Mesoplodon bidens, Ziphius cavirostris, Lagenorhynchus acutus, L. albirostris, Grampus griseus, Tursiops truncatus, Stenella coeruleoalba, Delphinus delphis, Pseudorca crassidens* (*Artsdatabanken, 2021*; *Kovacs & Lydersen, 2006*), Denmark and Greenland.

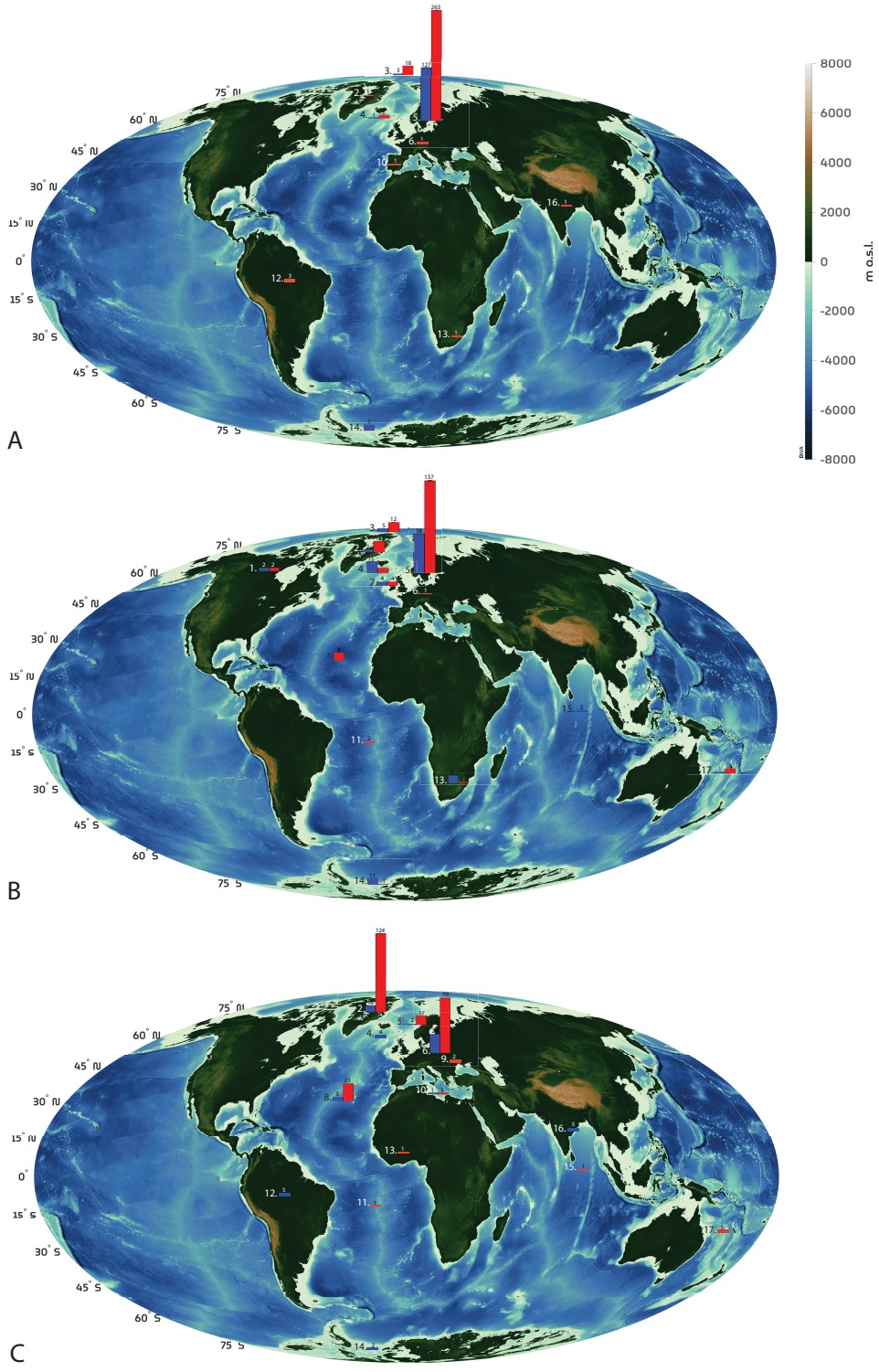

**Figure 4 Geographical representation of the whale specimens in NHM Bergen (A), NHM Oslo (B) and NHM Copenhagen (C).** Note that these are the specimens with a recorded country or ocean of origin. Specimens recorded from inland Norway, "Africa" and "Pacific Ocean" were not included. Otherwise the geographical locations are used as in the dataset, even when given with different levels of precision. Legend for areas of origin: 1 Canada, 2 Greenland, 3 "Arctic Ocean", Svalbard and Jan Mayen, Arctic Ocean, 4 Iceland, 5 Norway, 6 Denmark, 7 UK including the Hebrides, 8 North Atlantic Ocean

**Figure 4** (continued)
including the Faroe Islands and Azores, 9 North Europe including Sweden and Germany, 10 Mediterranean Ocean including France and Spain, 11 South Atlantic, including St. Helena specimens only, 12 South America including Brazil, Venezuela, Argentina and Puerto Rico, 13 Southern African countries of South Africa, Namibia, Angola, and Senegal, 14 South Georgia and Falkland Islands, and "Antarctic", 15 The Indian Ocean, 16 India and South East Asia including Sri Lanka and Thailand, 17 Australia, New Zealand and Solomon Islands. Map by Fabio Crameri. This graphic by Fabio Crameri adjusted from *Crameri, Shephard & Heron (2020)* is available *via* the open-access s-Ink repository. Red represents toothed whale specimens, blue represents baleen whale specimens.

# RESULTS

## Characterization of the collections

### The University Museum in Bergen

At NHM Bergen, the collection of whales started in 1834 (*Aslaksen, 2020*; *Kalland, 2014*). This study shows that most of the whale specimens were collected between 1880 and 1920 (Fig. 2), similar to other zoological specimens in the same collection (*University Museum of Bergen, 2023*). After 1920, more toothed whales than baleen whales were collected; 21 percent of the toothed whales and 13 percent of the baleen whales out of the whole collection (Fig. 2A).

The collection houses 24 whale species, out of which 159 specimens are baleen and 347 toothed whales (Fig. 5A). The most common species is harbour porpoise (*Phocoena phocoena*, 127 specimens) followed by minke whale (*Balaenoptera acutorostrata*, 113). Eighteen specimens are not identified to species level. Among species occurring in Norway, *Grampus griseus* and *Tursiops truncatus* are not represented (Figs. 2 and 5). Nearly all specimens of species that occur in Norway were collected in Norway (Fig. 4A). Only a few specimens in the collection are species not occurring in Norway: *Platanista gangetica*, *Inia geoffrensis*, *Sotalia fluviatilis* and *Steno bredaensis* (one specimen each) (Fig. 5A).

Only 70 (14%) specimens have some information about the collector, and 87 (17%) specimens on how the museum acquired them. However, it is known that many were bought from people hunting toothed whales in the areas around Bergen in the last half of the 19[th] century (*Aslaksen, 2020*; *Kalland, 2014*). This is confirmed by data assembled for this study (Fig. 4A); when acquisition mode is known, specimens bought by the museum are most common, and all of these, except one sperm whale, are toothed whales from Norway, collected before 1910 (Fig. 2). The purchase of specimens from whaling resulted in many wet collection foetuses, and skeletons (Figs. 1B and 1C). Among the 509 whale specimens, more than half (58%) are foetuses preserved in ethanol or formalin, and 31% are osteological specimens. The remaining specimens are organs, preserved in the wet (6%) or dry (2%) collection; and baleen (2%) (Fig. 2). The most common organs preserved are pieces of skin and dorsal fins, but there are also reproductive, digestive system and sensory organs.

Twenty-one specimens are registered as gifts to the museum, but it is assumed that this is true for many more. As the museum actively asked for donations of whale specimens and cooperated with the industry and the public (*Kalland, 2014*). People with a job in the marine sector are the most common donors, but there is also one author, one schoolboy,

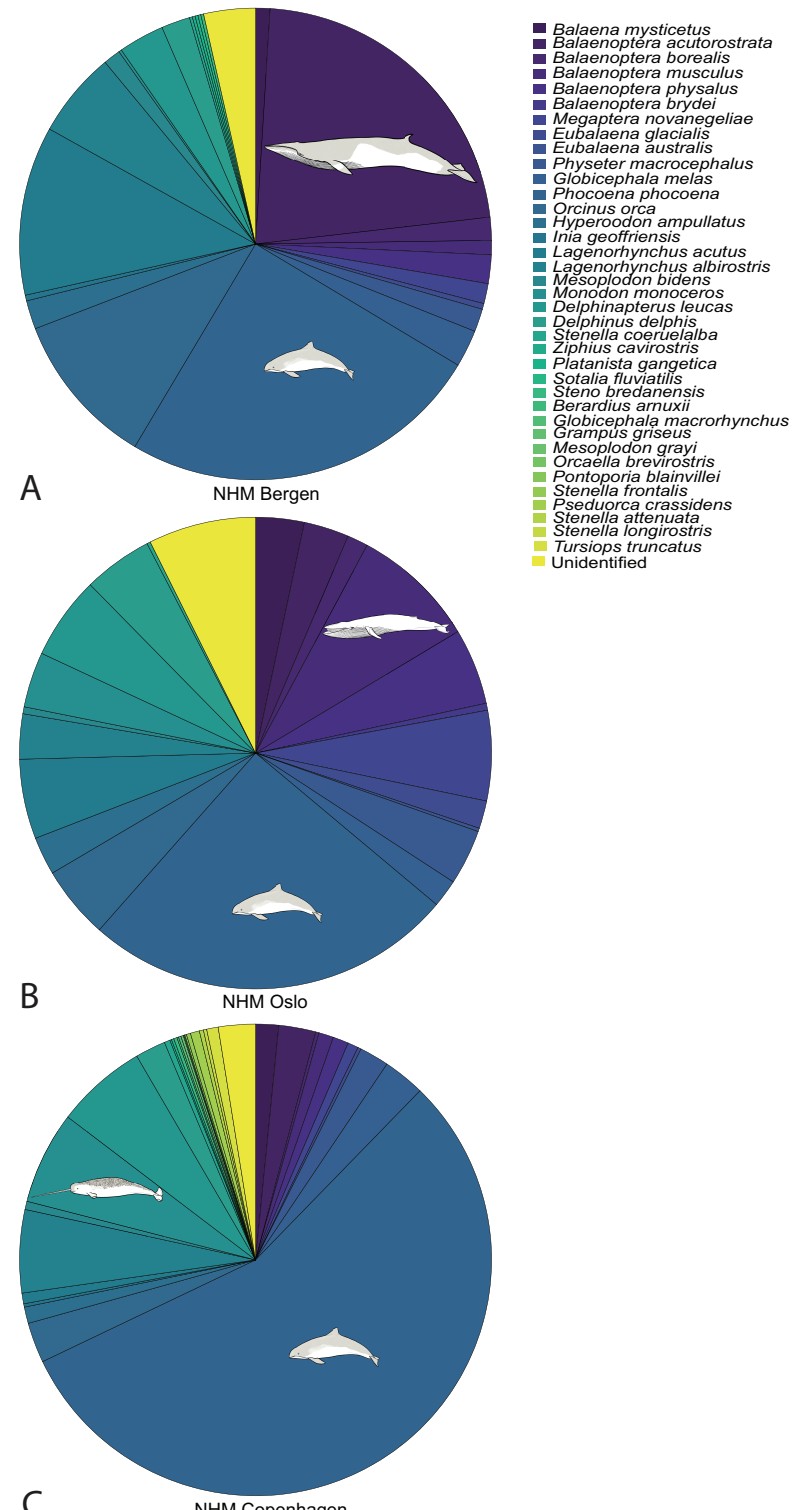

**Figure 5 Species composition for the collections in NHM Bergen (A), NHM Oslo (B) and NHM Copenhagen (C).** Drawings show the two most common species in each collection. Whale drawings: Nicola Dahle.

two businessmen and one kindergarten. There has always been extensive interaction with other institutions, such as the Institute for Marine Research, the local aquarium, the whaling museums in Sandefjord (Hvalfangstmuseet) and Tønsberg (Slottsfjellmuseet), and the natural history museums in Oslo and Copenhagen.

Eight specimens were sourced from Southern hemisphere whaling, mostly from South Georgia and one (a 1914 sperm whale) from South Africa. Three specimens are from 1913, and four from 1948. Six are foetuses, one is a set of hypophyses of minke whale (*Balaenoptera acutorostrata*), all in the wet collection.

## The natural history museum in Oslo

The first collected whale specimen was probably an orca killed in the Oslo fjord in 1820 (NHMO-DMA-25160/1-O). It could not be confidently located in 2022 but might be one of the unlabelled specimens. This study shows that most specimens that are still in the collection, came to the museum between 1860 and 1910 (Fig. 3A), like other parts of the zoological collections (*Johannessen et al., 2023*; *Johannessen & Lifjeld, 2022*). There is also an intake spike in 1999–2000, which is specimens of *Phocoena phocoena* for a research project (Ø. Wiig, 2023, personal communication).

Today, three quarters of the specimens are osteological, 15% are foetuses/juveniles and 10% are organs in the wet collection, whereas 1.5% are baleen (Fig. 3B). Twenty different species are represented (Fig. 5B). Like the collection in NHM Bergen, harbour porpoise (*Phocoena phocoena*, 107 specimens) is the most common, for which the majority are osteological specimens, as well as some foetuses and inner organs. The second most common is blue whale (*Balaenoptera musculus*, 36 specimens). Note that the latter are not complete osteological specimens, but rather disarticulated osteological elements, foetuses, and a few inner organs in the wet collection.

Compared to NHM Bergen, there is a larger geographical spread, with more specimens from other countries (Fig. 4B). For species occurring in Norway, NHM Oslo misses *Grampus griseus, Tursiops truncatus* and *Ziphius cavirostris*. Among the species occurring in Norwegian waters, most specimens originate from Norway, except the sperm whale (*Physeter macrocephalus*), for which the specimens originate from the UK, Iceland, and Saint Helena. Among species not occurring in Norwegian waters, NHM Oslo holds one *Eubalaena australis* and one *Balaenoptera brydei* specimen (Figs. 4B and 5B).

For approximately half of the specimens (189), the collector is known. Early in its history, the museum bought many specimens and received many as gifts, mainly from people working in the marine sector, but also citizens in Southern Norway who found stranded whales. An important source for specimens were the active measures taken by Professor Robert Collett, who travelled in northern Norway and cooperated with the whaling companies, including the famous whaler Svend Foyn.

Twenty-one specimens originate from the Southern Hemisphere and likely result from whaling (Fig. 4B). Most of these are humpback whales *Megaptera novaeangliae* (nine specimens), whereas the rest are large baleen whales, two sperm whales and two *Delphinus delphis*. They reflect the international Norwegian whaling; most are from South Georgia, and some from Angola, Namibia, South Africa, and the Kerguelen Islands.

### The natural history museum in Copenhagen

This is the largest osteological collection of whale specimens, with larger variation in species composition than the two Norwegian ones. More than half of the specimens are harbour porpoise (*Phocoena phocoena*, 988 specimens), followed by narwhal (*Monodon monoceros*, 113), beluga (*Delphinapterus leucas*, 111) and white-beaked dolphin (*Lagenorhynchus albirostris*, 101) (Fig. 5C). There are 136 baleen whale specimens, among them complete or close to complete skeletons of blue, sei, minke, bowhead and humpback whales (Fig. 1A).

Data on collection year are limited (Table 1), but indicates that the specimens in the osteological collections were collected from 1838 to 2017. Because so few specimens have a known collection year, temporal trends are not discussed. The collection includes all species that occur in the waters of both Greenland and Denmark. The *Eubalaena glacialis* specimens originate from Iceland and Spain, and the *Tursiops truncatus* from Azores and Faroe Islands. Marine species not occurring in either Denmark, Faroe Islands or Greenland waters in the collection are *Berardius arnuxii* (two specimens), *Mesoplodon grayi* (1), *Globicephala macrorhynchus* (three specimens, Senegal) and *Stenella longirostris* (four specimens, Australia). There are also coastal *Orcaella brevirostris* (Thailand, one specimen) and *Pontoporia blainvillei* (six specimens, Argentina), as well as freshwater toothed whales *Platanista gangetica* (India, three specimens) and *Inia geoffrensis* (four specimens, Venezuela) (Fig. 5C).

## DISCUSSION

This is the first study of the collection history and possible resulting biases in the three whale collections at the natural history museums in Bergen, Oslo and Copenhagen. Despite not being the largest whale collections worldwide, it can be argued that they are important because of the long coastlines, Norwegian whaling history and the Danish colonial history. The collections include specimens that are rare in a global context, and there is a potential for future knowledge of whales and their biology. In addition to investigating the patterns of whale collection and their drivers, this study asked how whaling has impacted natural history museum collections, and found that the impact is significant, however not from the period where most whales were caught.

Temporal patterns illuminate how the collection processes are closely related to society and history (Figs. 2 and 3). The major influx of whale specimens to the collections happened from the 1860s to 1920s (Figs. 2 and 3), together with the rise of modern science, the establishment of natural history museums in many countries (*Farber, 1982*), and for Norway, with nation building and modern whaling. As a part of Norwegian nation building and because of the focus on science, museums were established; in 1813 the Natural Museum in Oslo, and in 1825 Bergen Musæum (*Aslaksen, 2020*; *Wiig & Bachmann, 2013*). In the latter, the exhibition in the "Whale hall" has been an attraction since its opening in 1865, with around 20 complete whale skeletons, including an iconic blue whale, caught in 1878 in Finnmark in Northern Norway (*University Museum of Bergen, 2011*).

The Natural History Museum in Copenhagen can be traced back to the 17th century Museum Wormianum. The Zoological Museum was established in 1862 by three institutions merging: the Royal Kunstkammer, The Royal Natural History Museum and the Zoological University Museum (*University of Copenhagen, 2023a*, *2023b*). For the collection in NHM Copenhagen, the colonial relationship to Greenland and the stranding programme, that started already in 1885, are the most important contributors (*Ijsseldijk et al., 2020*; *Kinze, 2023*). This shows that regulations by law are important, for instance on ownership of stranded whales, which was and is important for NHM Copenhagen, as for other museums around the world (*Kinze, 2023*; *Kinze, 2017*; *Lotzof, 2023*; *Yamada et al., 2006*). This contrasts the situation in Norway, which has not had a system for recording and collecting all stranded whales in modern times.

## The making of the collections

Whales enter museum collections in different ways, after being killed and sometimes exploited, or because of stranding. A museum acquires a specimen either because it is donated, purchased, exchanged for another specimen, or it enters as part of a research project (*Aslaksen, 2020*; *Bakker et al., 2020*). The museums studied here seldom received whales in exchange, but especially NHM Bergen has exchanged a large number of whales for other animal specimens (*Aslaksen, 2020*; *Kalland, 2014*), which is to some degree reflected in the acquisition books. The museum also sold whale specimens to other museums (*Torino & Nicola, 2013*). Sometimes the decision to include a specimen happens first, such as when a museum actively orders a specimen of a particular species; in other instances, the museum is offered a specimen, and can either accept or decline. Intake bias, *i.e.*, the reasons behind decisions about accepting or actively seeking and acquiring specimens, can be nature-induced due to abiotic factors, taphonomy and decay, or anthropogenic due to societal factors, technological possibilities and limitations, economy and trends in culture and science (*Whitaker & Kimmig, 2020*).

Both the opportunistic, *ad hoc* collection mode and the dependence on the collectors' interests is commonly seen in biological collections (*Ponder et al., 2001*; *Pyke & Ehrlich, 2010*).

A specimen often passes through at least one person before it enters a museum collection. Decisions as to who collected and who took the specimens, and their interest at a certain point in time, has largely influenced the resulting collections. Many people were involved: scientists, other museum employees, local industry, natural history traders and the public. The museum employees, in particular, were important; R. Collett in Oslo, D. F. Eschricht in Copenhagen, and W. Christie, F. Nansen and G. Guldberg in NHM Bergen actively collected, and ordered specimens, including from each other (C. Kinze, 2023, personal communication) (*Guldberg, 1885*; *Kalland, 2014*).

Greenland has unique access to Arctic cetaceans, and because it was a Danish colony, this is evident in the collection in NHM Copenhagen, as it has been for cultural heritage (*Gabriel, 2009*), and this is a common situation in European museums (*Bakker et al., 2020*). Several specimens in the collection are a result of D. F. Eschricht's cooperating with captain Holbøll, who organized for narwhal and beluga specimens to be sent to the museum

(C. Kinze, 2023, personal communication). Indigenous hunters and traders in Greenland were and are very important for the acquisition of specimens, but none of their names are present on the specimen labels, which means they receive less recognition than the Danish people involved.

Some specimens result from expeditions, which is typical for the 1800s and 1900s and still is today (*Heyning, 2002*). At NHM Oslo this includes *e.g.*, a foetal narwal skull from Jan Mayen, donated by Roald Amundsen, from the first Gjøa expedition in 1901, a *Delphinus delphis* specimen from Australia from Carl Lumholtz in 1880, and two *Balaenoptera* specimens from South Africa 1912–1914 from Ørjan Olsen (see Supplemental Material). At NHM Copenhagen, one specimen (*Platanista gangetica*) resulted from the first, and four (*Delphinus, Stenella* and *Globicephala*) from the second Galathea expedition (*Bruun, 1957*).

Collecting whales can also be a logistical challenge because of their size and smell (*Heyning, 2002*; *Pyenson, 2017*). Whale specimens are preserved in different ways (Figs. 1, 2B and 3B). This is a human decision, made at an intersection of wanting to preserve as much information as possible, research trends, but also what is logistically and economically possible. Whales have not only come into the collections, they have also left; through discarding due to decay or space limitations, through exchange with other museums or due to random events such as fire or flooding (*Pyke & Ehrlich, 2010*).

Today, fewer whale specimens enter the collections in the three NHMs. This is probably due to a combination of research focus, economy, limited storage space and increasingly stricter regulations on hunt and transport of animal specimens. The three NHMs still accept some donations and sometimes collect stranded specimens, especially if they complement the existing collection (*University of Bergen, 2023*) and there are examples of recent collecting for research purposes (T. Lislevand, 2022, Ø. Wiig, 2022, personal communication). At NHM Copenhagen, the stranding programme has made available a number of specimens in much more recent times than at the other two NHMs (*Kinze, 2017*).

## The effect of whaling

Many specimens in the museum collections result from whaling, such as NHM Bergen, where many specimens come from local hunt on toothed whales since the mid-1800's, which was more opportunistic than the hunt for baleen whales (*Collett, 1911–1912*; *Kalland, 2014*). For instance, in 1885, a group of approx. 1,000 *Lagenorhynchus acutus* assembled in a fjord close to Bergen, out of which 200–300 were killed (*Collett, 1911–1912*; *Rasch, 1845*). Then curator at NHM Bergen, Fridtjof Nansen, arranged for the museum to buy foetuses and skeletons (*Collett, 1911–1912*), out of which approx. 20 are still part of the collection (Fig. 2, Supplemental Material). Whaling was also the method for acquiring large baleen whale specimens from northern Norway, both for NHM Bergen and NHM Oslo.

Surprisingly few specimens from the Southern Ocean made it into the Norwegian collections. Some were bought and some were donated, but for most of them, acquisition mode is unknown. There are more Southern Ocean specimens at NHM Oslo

(21 specimens) than at NHM Bergen (eight specimens), reflecting the more diverse geographical scope of the collections at the museum in the capital (Fig. 4).

Why did so few specimens enter the collections from the largest scale whale hunt? It might result from inherent factors in the industry, or the time when this happened. In the 1900s, biological sciences moved away from specimen-based natural history to studies on ecosystems and molecular studies (*Burnett, 2012*; *Farber, 1982*; *Gippoliti et al., 2014*). Many museums also experienced limited space for growing collections (*von Achen, 2019*). The set-up of the whaling industry itself was targeted for commercial purposes, and had a long distance to Norway, and gradually changed to pelagic factory ships. The large-scale whaling in the Southern Ocean was a sheer industrial endeavour, which did not grow the collections in Oslo and Bergen. There is however one exception: The collections at the Anatomical institute in Oslo, that through cooperation with whaling ship medical doctors, received approximately 300 whale brains and foetuses, which were used for neurobiological comparative studies (*Dietrichs, 2018*; *Jansen & Osen, 1984*). The collection has recently moved to NHM Oslo but is not part of this study.

## Bias in the collections

Whenever the collections deviate from nature, there might be two types of biases at work that are intertwined: natural- and human-induced. The three collections of whales are the result of whale populations in the oceans being filtered through all of these.

The species composition in all three collections show the same trend; local and common species are abundant, in addition to the strong colonial bias towards *e.g.*, Arctic species from Greenland in NHM Copenhagen (Fig. 5C).

In all the three collections, as in nature, there are more toothed whales than baleen whales. The harbour porpoise is an example, as it is one of the most common species in coastal waters as well as in the collections (Fig. 5). With regard to baleen whales, how common a species is, can also influence collecting. At NHM Bergen, 71% (113 specimens) of the baleen whale specimens belong to one species, the minke whale (*Balaenoptera acutorostrata)*. The reason is probably a combination of natural abundance and human factors; minke whales are the most common baleen whales and have been hunted for a long time, which means that more specimens are available for the museums. Lastly, the few whale species that do occur in Norwegian waters but not in the collections, are species that are uncommon and often do not reproduce in Norway.

Which whale species are common has changed significantly through the years in which these collections have been assembled, because of whaling (*Rocha, Clapham & Ivashchenko, 2014*). In modern whaling, the largest and more coastal species were usually hunted first, and this is reflected in the collections. From the years after the ban on whaling in Norway (1904), very few baleen whales were collected (except minke whale) and those who were, originate from the Southern Ocean. One example is the North Atlantic right whale *Eubalaena glacialis*, where no specimens entered NHM Oslo or NHM Bergen after 1904. Today, the species is regionally extinct in Norway and a critically endangered species worldwide (*Artsdatabanken, 2021*).

The "local" species composition is still changing. One resulting bias in NHM Oslo and NHM Bergen caused by timing, is the lack of *Tursiops truncatus* specimens. This species used to be uncommon, but is increasingly more common today because of increased sea temperatures (*Artsdatabanken, 2021*). The same is true for *Mesoplodon bidens* and *Ziphius cavirostris* in Denmark. The latter had its first stranding in 2020 and was added to the museum collection, which previously had a specimen from New Zealand (*Alstrup et al., 2021*; *Stavenow et al., 2022*). Non-native and invasive species are often overlooked in collecting (*McLean et al., 2016*). The number of whale strandings might increase in the future, because some populations are growing, but it has also been argued that anthropogenic pressures might induce strandings (*Aniceto et al., 2021*; *Ijsseldijk et al., 2020*; *Stavenow et al., 2022*).

Among the few non-local specimens, freshwater toothed whales (*e.g., Platanista gangetica* and *Inia geoffrensis)* are more common than coastal or open ocean species. A possible reason is that these were "exotic" and thus interesting either for comparison to local species or for exhibition purposes (*Bakker et al., 2020*). This shows that there can be biases in both directions; both favouring the common, as above, whereas in other instances focusing on the uncommon.

The one *Platanista gangetica* specimen (not located) in NHM Bergen was received from the Ganges, India, from G. A. Frank in 1898. Frank was a natural history dealer in Amsterdam with a large global network, and one of museum's most important trade partners (*Kalland, 2014*; *Largen, 1985*). He also traded *P. gangetica* specimens to the natural history museums in Leiden and Pisa, and to the former, whale specimens from Norway *e.g.,* two *Lagenorhynchus* specimens (*Braschi, Cagnolaro & Nicolosi, 2007*; *Broekema, 1983*). Such trade networks were common and important for the museums (*Coote et al., 2017*), and this indicates that NHM Bergen actively wanted a *Platanista* specimen. The *P. gangetica* specimens in NHM Copenhagen were collected 1840–1845 by a Dr. Mundt, and by the first Galathea expedition.

NHM Bergen has one complete *Inia geoffrensis* skeleton, collected in 1924, from Manacapuru, Brazil. W. Ehrhardt is the collector, probably the German taxidermist and collector who supplied museums with vertebrates from Brazil (*Gutsche et al., 2007*). There are four *I. geoffrensis* specimens at NHM Copenhagen. Two of the skulls were collected in 1892 in Rio Apure, Venezuela, by van Dockum, probably the captain in the Danish fleet, on trips to the colony "Danish West Indies" (Islands St. Thomas, St. Jan and St. Croix) (*Garde, 1952*). There is also an almost complete skeleton of *Sotalia fluviatilis* in NHM Bergen (BM 414) from Rio de Janeiro, Brazil, given by professor van Beneden. *van Beneden (1864)* is the author of *Sotalia guianensis*, but this specimen might have been collected by his son Edourd van Beneden during his travels to Brazil in 1872.

One example is the differing blubber amount among whale species, which means that some whales float after death whereas others sink (*e.g.,* thick blubber in bowhead whales means they float). This influences stranding potential of different species and is thus a natural-induced bias, acting together with ecology and behaviour of the species. The blubber content however also influences whether humans hunt certain species,

showcased by the invention of modern whaling where rorquals became more easily available for hunt and thus for museum collections (*Collett, 1911–1912*).

Correct cataloguing, detailed acquisition records and labelling of museum specimens is crucial for later use and analyses of historical specimens, but missing data was the case for many specimens in this study, hindering insights, especially on bias (*Lane, 1996*; *Pyke & Ehrlich, 2010*). This is evident in this study by the lack of information for many whale specimens, especially on ontogeny and sex (Table 1). The large number of foetuses in NHM Bergen might result from the large number of available specimens from whaling, but there are also historical reasons. D. F. Eschricht quoted Georges Cuvier about whales being too large to be preserved completely. *Eschricht (1844)* argues that foetuses are important study subjects that are not interesting for other people (because of limited economical value), adding that collecting foetuses make possible to study the entire anatomy of the whale, only on a miniaturized scale.

Sex bias is also common in museum collections, with male specimens being more commonly collected among mammals and birds (*Cooper et al., 2019*). Many specimens are needed to understand sexual dimorphism, and not knowing sex can lead to biased and incomplete results in *e.g.*, taxonomy, comparative anatomy, genomics and toxicology (*Cooper et al., 2019*; *Heyning, 2002*). In the three collections, sex is known for too few specimens to infer any collection-wide trend (Table 1). A possible bias that is sex-related is that narwhal specimens with a tooth are usually interpreted as male, even if some females also develop a tusk, and some males can be toothless (*Petersen et al., 2012*).

## Future research

The three collections have contributed to research in the past, and can provide valuable data also for future research and outreach. From the early phases of the collections, it is worth noting the extensive work by Eschricht using whale specimens in Copenhagen for research on anatomy *e.g.*, (*Eschricht, 1844*, *1846*), Guldberg and Grieg's research on whale foetuses and embryology using the collection in Bergen *e.g.*, (*Grieg, 1898*; *Guldberg, 1894*; *Guldberg & Nansen, 1894*), and the description of a possible new species (today a syntype) by Halvor Heyerdahl Rasch in Oslo (*Rasch, 1845*). There has been a relationship between research and whaling, notably on population structure and reproduction, even if this has not always been conducted in the natural history museums (*Burnett, 2012*). A British tax on whale oil supported the series of voyages called the "Discovery investigations", aimed to understand Antarctic marine biology, amongst it whale populations and life history. The South Georgia Marine Biological Station, established in 1925, was also funded by the Discovery investigations (*Burnett, 2012*; *Survey, 2017*). More recent research using the collection specimens include skeletal evolution and adaptations to diving and feeding underwater *e.g.* (*Delsett et al., 2023*; *Galatius et al., 2020*, *2009*) but also strandings (*Christian Kinze et al., 2021*; *Ijsseldijk et al., 2020*).

The future use of museum specimens often cannot be predicted, because of the emergence of new scientific questions and technologies. However, based on the current situation, some directions can be suggested based on what the three collections hold. In the two Norwegian collections the large number of foetuses preserved in ethanol or formalin is

noteworthy. These specimens can contribute to the understanding of whale embryology and development, and from that, life history parameters, which are lacking for many species and might enable *e.g.*, studies of soft tissue (*Heyning, 2002*; *Lotzof, 2023*; *Yamato & Pyenson, 2015*; *Gavazzi et al., 2023*; *Martin & Da Silva, 2018*). Other notable specimens are those from the large baleen whales, both including skeletons and some specimens with inner organs. For some of the smaller toothed whales, the large quantity (Fig. 5) might also provide a valuable resource. The age of many specimens, up to 150 years old, can also be valuable for research on changing composition of sea waters and concentrations of environmental toxins. Geographical information that can be gained from collections are often limited because they are often collected opportunistically and based on the collector's interests (*Bakker et al., 2020*; *Pyke & Ehrlich, 2010*), which is also likely the case here.

Several other museums in the two countries have whale collections, with fewer specimens than the three studied here. In Norway this include the university museums in Trondheim and Tromsø, as well as the whaling museums in Sandefjord and Tønsberg. The latter two have complete skeletons of many species on display, including most of the large baleen whale species. Apart from these, the extensive Norwegian whaling is not a part of museum exhibitions, which might in itself be an interesting topic for future research.

Future collection will be different from past collection of whale specimens, for reasons outlined above due to whaling, research trends and regulations, but also because of the collectors. This work exemplified the role of Indigenous people of Greenland, which is underacknowledged. The number of women among the collectors in the dataset is also close to zero. This is important because women are often underacknowledged for their scientific contributions (*Meeus et al., 2021*), but also because collectors of different genders might favour different objects in their collecting (*Cooper et al., 2019*). It is increasingly acknowledged by museums that collection history and who the collectors are, influence and might bias the collections, and mitigating actions can be taken.

## CONCLUSION

This work is based on studies of three collections as they were in 2022. The combination and interaction of natural and anthropogenic factors in the collection processes have formed the collections as they are today and will continue to do so in the future (*Anderson & Pietsch, 1997*; *Whitaker & Kimmig, 2020*). The whale specimens are not a 1:1 representation of whale populations, but instead reflect the collecting biases of the people responsible, combined with biological factors such as stranding potential, commonness and buoyancy, and cultural factors such as research trends and economy. The NHM Bergen collection is biased towards local species that were hunted, and in favour of complete foetuses. At NHM Oslo, the research focus is less clear, and the collection is instead assembled more at random, including international specimens and some specimens from Southern Ocean whaling. NHM Copenhagen has two major biases; that of the common harbour porpoise from Danish waters, and narwhals and belugas from the former colony Greenland in the Arctic.

Natural history museum collections played a role in nation building projects as educational and research institutions, that took place in the era of industrial whaling. Compared to Norway's extensive whaling, the number of specimens from the Southern Ocean is surprisingly small. It seems that the actions taken by the museum itself are more important for the resulting collections than large scale industrial trends. A very important factor is thus the museum employees involved in decision-making and their collaboration with the industry, public and traders, and their documentation of the specimens and their collection history. The colonial history is also clearly visible, especially in the large number of Arctic specimens such as narwhals, held at the NHM in Copenhagen. Together this again emphasizes the value of well contextualized specimens.

The Natural History Museum whale collections in Oslo, Bergen, and Copenhagen document an important animal group in the oceans. They have been and will continue to be important for research, especially in a time of environmental change for which museum collections can contribute to long time series, provide specimens of endangered species, as well as historical and cultural background. This work also show that documenting the collection history and recording biological information about specimens greatly enhances the research potential. By focusing on collection history and possible biases, this work contributes to the continuous building of knowledge derived from the specimens. Together, this can contribute to ensuring the afterlives of the whales (*Nicolov, 2019*; *Vieira et al., 2020*) The increased focus on museum specimens hopefully can result both in important science and in the long-time management of these unique objects, which is sometimes lacking resources (*Bakker et al., 2020*; *Boakes et al., 2010*; *Gippoliti et al., 2014*; *Vane-Wright & Cranston, 1992*).

## ACKNOWLEDGEMENTS

Research assistant Nicola Dahle is thanked for the whale drawings and for entering some of the data into the datasets. The author also warmly thanks the museum curators and employees that gave access to collections and answered questions: H. Meijer, T. Lislevand, A. K. Hufthammer, J. Magnussen and B. R. Olsson at NHM Bergen; L. E. Johannessen, K. L. Voje, B. Lund at NHM Oslo, and P. R. Møller, L. I. Ahl, L. Cotton; B. Lindow at NHM Copenhagen. Ø. Wiig and Carl Kinze provided very valuable information on collection history. K. Osen and E. Dietrichs are thanked for discussions about the Anatomical institute whale specimens. U. Spring, J. Kaasa, A. Simon-Ekeland, S. Gaupseth and J. Schimanski are thanked for good discussions and feedback. H. A. Nakrem gave valuable feedback on an earlier version of this manuscript.

The editor and two reviewers are warmly thanked for very valuable comments that greatly improved the manuscript.

### Funding

This is a Collecting Norden article, funded by UiO Norden. The collection visit in Copenhagen was supported by a Synthesys+ grant, and a visit to the NTNU

Vitenskapsmuseet by a Professor R. Collett og Professor N. Willes legat research grant. The funders had no role in study design, data collection and analysis, decision to publish, or preparation of the manuscript.

## Grant Disclosures
The following grant information was disclosed by the authors:
UiO Norden.
Synthesys+ transnational access grant.
Professor R. Collett og Professor N. Willes legat research grant.

## Competing Interests
The author declare that there is no competing interests.

## Author Contributions
- Lene Liebe Delsett conceived and designed the experiments, performed the experiments, analyzed the data, prepared figures and/or tables, authored or reviewed drafts of the article, performed all collection visits and took pictures used in Fig. 1, and approved the final draft.

## Data Availability
The raw data is available in the Supplemental File.

## Supplemental Information
Supplemental information for this article can be found online at http://dx.doi.org/10.7717/peerj.16794#supplemental-information.

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
