# Peer review of "Collecting whales: processes and biases in Nordic museum collections"

_PeerJ, doi:10.7717/peerj.16794_

## Round 0.1 · original submission · Major Revisions

Both reviewers are positive about this interesting submission and, although they have made comments/extensive comments on the manuscript, they (and I) encourage you to submit a modified version of the work. In particular, it would be good to see a representation of the geographic sources of the collections (basically a map). Although the reviewers both suggested "minor revisions" I think that in total the changes they see as necessary are quite significant and I would like to send a revised version out for another review.

There are minor issues with language throughout that they and I have tried to assist you with.

Reviewer 1 ·

Basic reporting

The text as a whole could do with some streamlining to tighten it up. There are some issues with tenses, words missing, spelling mistakes that need fixing (see pdf comments) but overall it is well written. Colloquial language is a problem in a few places.

Is table format in journal style?

Figure captions need more information and explanation, some figures need scale bars.

(see also comments on attached pdf)

Experimental design

Introduction (see also comments on attached pdf):

L51-55. Can you more clearly define what you’re referring to in this paragraph? I’m not sure that it’s true to say “most people probably encounter whales in natural history museums”. There are other ways that people can engage with whales, such as TV documentaries for example, that are probably more frequent occurrences than museum visits. While controversial, have you considered cetaceans in captivity as a way in which people might come across whales in the modern age?

Para starting on line 57. It would be helpful to me to know exactly what you mean when you discuss whales in museums, and what the scope of this study is in reference to that. I can think of many examples in which whales might be represented in collections, such as physical remnants in social history collections, fine art and folk art, architecture etc. as well as whaling-related ephemera. Perhaps you could be specific about the fact that your study only looks at biological specimens in natural history collections, but perhaps also acknowledge that whales and related objects appear across the spectrum of museum collections. A very brief overview of how biological specimens are presented (i.e. what a researcher might be likely to find) might be helpful at this stage.

L67-68. Can you expand on this to explain and give context to collections bias in more general terms? It is a phenomenon that exists in all museum collections and I think warrants greater depth for uninitiated readers. You expand on causes (e.g. loose collections policies, socio-cultural bias inherent at the time of acquisition, collecting priorities of individual curators, availability of suitable materials etc.) and potential impacts for researchers. Museums are also now more aware of this as an issue so are putting mitigating processes in place such as regularly reviewed collecting policies and acquisition by committee.

L84-85 “A review of the previous and future research based on the specimen in this collection is outside the scope of this research”, I don’t think it is out of the scope, because including examples (rather than an exhaustive review) would help readers and potential users of the material and database to see how they might be used and to understand how they have been used previously. This can provide a summary of the opportunities and potential, I believe this to be important given the papers aims.

Lines 99 -104. I’m not sure how the two halves of this paragraph link to each other. Make the connection explicit.

Methods:

Better documentation of the process and description of the collections is required. The methods needs restructuring to follow a logical order and remove unnecessary repetition, there is also key information missing. What databases In particular did you use to assemble this database? Due credit should be given to those who assembled the original databases.

Was any visual documentation of the collections done? Or is it available from the museums? Given that there were some major issues with labelling this might be a useful endeavour.

Validity of the findings

Results (see also comments on attached pdf):

Needs a table summarising specimens and which collections plus labelling/mislabelling, missing information and any missing specimens etc. to support the descriptions on L171-222

I would like a visual summary of the spatial information for the specimens from each collection present on a map or other figure (or table )– map with points and dates perhaps? To allow us to compare the distribution etc.

Is table format in journal style?

Figure captions need more information and explanation, some figures need scale bars.

Discussion:

The opening paragraph needs some work to recap aims and relevance, then main result.

The collections history needs a little more detail and info on how it was researched. For instance, the intake journal is mentioned for one collection but not much else is said about this. There is a history of the museums generally in the introduction (L88-104), might this be better in the discussion? Or at least some mention in relation to bias as the collections history is relevant here.

L335 and 552 I would also like to see more discussion and suggestions of the potential uses of this material with citation of work that has done so in the past. Examples would help to demonstrate and therefore facilitate use of the collections. Additionally, it would show the value of this work and the published database.

The paper highlights some specimens of importance in the section on biases, but I would like to see examples of what the important specimens are and how they might be used.

L373-376. I’m really pleased to see this in here. A diversity of collectors and sources of material is important to help counter bias in collections. For example, male and female curators are likely to favour different acquisitions (Cooper et al, 2019 - https://doi.org/10.1098/rspb.2019.2025 - suggest that sex biases in natural history collections could be partly due to cultural attitudes to sex over time.)

L386-390. I don’t know the answer to this, but are there legislative or ethical restrictions to museums acquiring particular species?

L457 and elsewhere. A summary of collection bias is needed in the discussion/conclusions and the implications of this in terms of research potential. I would like to see more consideration of biases from museum collections as a whole as well as specific to whales/large mammals (with reference to existing literature) and these collections specifically. If your aim is to highlight the collections and encourage reuse then this is needed.

L468 I would like to know what is the broader context on whale collections, what other important museum materials exist, whaling museums are mentioned for instance, where are they and how might they add to the picture from the collections described in this paper?

L495 for the non-museum community how are decisions around collections made, what are the reasons behind decisions about accepting or actively seeking and acquiring out specimens? – this will help to understand what the biases are. See also comments above on L373-376, 386-390

Conclusions:

Whale specimens in museum collections cannot be assumed to be representative of population numbers and distribution at the time of acquisition, but are more likely to reflect the collecting biases of those people responsible for collecting the material, combined with the various biological and cultural factors mentioned that might influence those decisions. At L71-72, the process is likened to filtration – perhaps reiterate this?

Can you support this argument by making a comparison with historical ecological data (which is clearly also flawed) to the whale types and quantities represented in museum collections? Are the distributions within collections comparative with known population density and distribution at the time of acquisition?

L539 Need to be clearer and more explicit about the biases.
L540, 552 what can these collections and data be used for? Reiterate
L543 could also argue importance of collection because of Norwegian whaling history

Additional comments

The paper is quite descriptive but provides a good opportunity to gain insights into how and why a researcher might use a collection. Its is also valuable to highlight the issues of bias within museum collections as an issue for future researchers. This work could be beneficial both to those wanting to use the content of these collections for research purposes, as well as being informative for those creating and managing those collections.

Having said this it could better inform on what the collections described contain, how they could be used and what the biases are with this information/samples.

I do find it a valuable piece of work if the suggested changes can be implemented.

I do wonder if there is another format that might be more appropriate? I can’t see it listed but assume its an original research article, but perhaps it should be a “data report”?

Annotated reviews are not available for download in order to protect the identity of reviewers who chose to remain anonymous.

·

Basic reporting

I did like the paper a lot, it is very interesting and overall of good quality and interest to peers and wider audiences. I think it can be accepted with minor revisions. There are some aspects that I think can be more problematic, but I added comments on those across the manuscript. The discussion might need some rearrangement.

Experimental design

No comment

Validity of the findings

No comment

Additional comments

No comment

---

## Round 0.2 · accepted · Accept

Congratulations on a job well done. Both the reviewer and I think this paper is good to go and you have done a great job of responding to their comments.

God Jul och Gott Nytt År !

There are a couple of really minor typos that should be picked up in the proofing stage.
52 are not rather than do not
86 straightforward rather than straight-forward?
188 head technician rather than head engineer?
246 missing comma after B. borealis
288 as the rather than asthe

Reviewer 1 ·

Basic reporting

I spotted a couple of typos - highlighted in yellow in the attached pdf.

Experimental design

Good

Validity of the findings

Good

Additional comments

I find this manuscript much improved, it is easier to read and many points have been clarified.

The structure works much better and highlights the value added by this contribution - this makes it more engaging and more useful to other workers.


The context that has been added ion the collections is very useful (and is fundamental to qualifying the biases). Additionally, I think consideration of bias is more complete and will be very useful to any readers thinking of working with whale collections.

I think the aded section of research potential is also very useful, for inspiration but also for promoting this work, these specific collections but also the use used of material collections as a whole.

As a final note I would say that this process of describing a collection and what it does and does not contain is illuminating in itself because it tells us something more about human behaviour e.g. of the collectors and the culture at the time they were collected (that is more nuanced than many other types of seemingly equivalent information).

Congratulations to the author on preparing an interesting and valuable manuscript.

Annotated reviews are not available for download in order to protect the identity of reviewers who chose to remain anonymous.